# Regulation of Dynamic Cell Adhesion by Integrin-Integrin Crosstalk

**DOI:** 10.3390/cells11101685

**Published:** 2022-05-19

**Authors:** Carl G. Gahmberg, Mikaela Grönholm, Sudarrshan Madhavan

**Affiliations:** 1Molecular and Integrative Biosciences Research Program, Faculty of Biological and Environmental Sciences, University of Helsinki, Viikinkaari 9 C, 00014 Helsinki, Finland; mikaela.gronholm@helsinki.fi (M.G.); sudarrshan.madhavan@helsinki.fi (S.M.); 2Drug Research Program, Faculty of Pharmacy, University of Helsinki, Viikinkaari 9 C, 00014 Helsinki, Finland

**Keywords:** integrin, cell adhesion, phosphorylation, crosstalk, transdominant regulation, SARS-CoV-2 receptor

## Abstract

Most cells express several integrins. The integrins are able to respond to various cellular functions and needs by modifying their own activation state, but in addition by their ability to regulate each other by activation or inhibition. This crosstalk or transdominant regulation is strictly controlled. The mechanisms resulting in integrin crosstalk are incompletely understood, but they often involve intracellular signalling routes also used by other cell surface receptors. Several studies show that the integrin cytoplasmic tails bind to a number of cytoskeletal and adaptor molecules in a regulated manner. Recent work has shown that phosphorylations of integrins and key intracellular molecules are of pivotal importance in integrin-cytoplasmic interactions, and these in turn affect integrin activity and crosstalk. The integrin β-chains play a central role in regulating crosstalk. In addition to Integrin-integrin crosstalk, crosstalk may also occur between integrins and related receptors, including other adhesion receptors, growth factor and SARS-CoV-2 receptors.

## 1. Overview of Cell Adhesion

Cell adhesion is pivotal for development, the generation of organs, cellular movement and cell distribution in the body. It engages adhesion proteins, which include integrins, immunoglobulin superfamily proteins, carbohydrate binding selectins and cadherins [1,2,3,4,5]. The integrins are glycoprotein heterodimers formed by an α-and a β-chain. They are type 1 membrane proteins with large extracellular domains, a transmembrane segment and a cytoplasmic tail. The integrins exist in at least three major forms (Figure 1). In the resting state, the integrin heads are turned towards the lipid membrane. Upon activation, the integrins extend and open the ligand binding sites. The integrins are activated by inside-out or outside-in activation, resulting in intracellular signalling and cell adhesion. In inside-out activation, an activating signal originates from a non-integrin receptor, and is transmitted to integrins. In outside-in activation, an extracellular ligand binds to an integrin and activates the integrin. Several cytoplasmic proteins interact with the integrin cytoplasmic tails and are important in regulating integrin activity. When activated, the integrins may increase their binding capacity by an increase in affinity of individual integrins through conformational changes, or by clustering of integrins, which is known as increased avidity.

Cells can adhere to a variety of cells, and to the extracellular matrix. To achieve this, they must express several different integrins with different binding specificities. These enable them to regulate adhesion according to their functional requirements. This is most obvious for mobile cells like leukocytes, which encounter different cells and extracellular molecules, for example when they leave the blood vessels and penetrate into tissues. These events require the activation of some integrins, but also the deactivation of others. Several studies show that integrins communicate between each other through intracellular signalling, and through such crosstalk the integrins are able to cooperate and direct cellular movement and adhesion in a useful way [6,7].

Furthermore, other receptors, like growth factor receptors and virus receptors, may communicate with integrins by crosstalk [8,9,10,11], but in this review we focus on integrin-integrin crosstalk. We first describe how integrins are regulated, and how they induce intracellular signalling. This information is necessary to understand how integrins communicate by crosstalk.

## 2. Cells Adhere through Integrins by Binding to Cellular and Extracellular Ligands after Activation

Integrins like α4β1 (VLA-4), α5β1 (VLA-5), αVβ1 and αIIbβ3 bind to RGD sequences found in several important extracellular matrix proteins such as fibronectin and fibrinogen, but also in some virus receptor binding proteins like the SARS-CoV-2 spike protein [12,13,14]. The leukocyte β2-integrin LFA-1 binds to ligands on other cells, to large domains in the intercellular adhesion molecule ligands (ICAM) [15], but also α4β1 binds to a large domain in the vascular cell adhesion molecule (VCAM-1) [16]. Recent work has shown that the bent LFA-1 form, facing the membrane and thought to be inactive, in fact can be labelled with the MAb24 antibody Fab fragment. This antibody is specific for fully activated LFA-1, indicating that the ligand binding site is open and the integrin is bound to an ICAM molecule in cis [17,18].

Integrins must be activated to be able to bind to their ligands. Inside-out activation takes place by signalling from a non-integrin molecule to an integrin. Well studied examples are signalling from the T cell receptor, and from chemokine receptors to LFA-1 [19,20]. Outside-in signalling occurs, for example, in platelets, when they become exposed to fibrinogen, which directly activates the αIIbβ3 integrin. Phorbol esters, which activate protein kinase C enzymes, are often used in studies on cell adhesion by inducing inside-out activation through phosphorylations [21].

## 3. The Integrin β-Chains Have a Pivotal Role in Integrin Regulation, Whereas Integrin α-Chains Are Important for the Formation of the Ligand Binding Sites

Both in inside-out and outside-in activation, the integrins are regulated by interactions of their intracellular domains with cytoplasmic proteins or adaptors. Proteomic analysis has identified a large number of proteins, which interact directly or indirectly with the integrins tails [22], but rather few seem to be absolutely required.

The cytoplasmic tails of the integrin β-chains contain three highly conserved sequence motifs, which are of pivotal importance in integrin adhesion and signalling. These are the two NXXY/F motifs, which flank a threonine/serine (T/S) rich motif (Figure 2). The proximal NXXY/F sequence acts as a binding site for talins, the T/S motif is part of binding site for kindlins, and the distal NXXY/F motif completes the kindlin binding site. FilaminA binding covers the three motifs. When phosphorylated, the 14-3-3ζ proteins bind to the T/S motif [23]. 

It is apparent that the integrin β-chains are essential in integrin regulation, and most well characterized adaptor proteins, which regulate adhesion and signalling, bind to the β-chains. On the other hand, the α-chains contain, or participate in forming the external ligand binding sites. Some important adaptors bind, however, to integrin α-chains including paxillin binding to α4 (see below). In β2-integrins, the α-chains contain an inserted I-domain (A-domain), which acts as the ligand binding site. In most integrins, the ligand binding site is formed by a combination of the α-and β-chain outermost domains [24].

To be able to understand how integrin crosstalk could take place, it is important to know how integrin induced adhesion and signalling is regulated. Integrin crosstalk is due to integrin signalling, but signalling does not mean that there is always crosstalk. Several reviews have been written on integrin signalling [2,5,25,26,27]. There is a large consensus that the cytoplasmic proteins filaminA, talin-1 and-2, the kindlins, the 14-3-3 proteins and α-actinin play an important role [28]. They all interact directly with the integrin β-chain cytoplasmic tails, and their binding sites have been mapped (Figure 2). In addition to these, there are also a number of other proteins interacting directly or indirectly with the integrin cytoplasmic domains, but their mechanism of action is less understood [22]. We should, however, be aware that the signals, coming through integrin inside-out or outside-in stimulation, must be able to initiate rapid and short lived changes in the molecular interactions between the integrins and the cytoplasmic proteins. 

By phosphorylation it is possible to rapidly change the interactions of the integrin cytoplasmic tails with cytoplasmic proteins, resulting in changes in integrin conformation and interaction with the cytoskeleton. In addition, phosphorylations can induce up-or downregulation of integrin affinity and avidity for their external ligands [5].

Importantly, cytoplasmic adaptors may also be phosphorylated, and for example kindlin-3 is phosphorylated on several sites, of which at least T482 and S484 are functionally important [29,30]. When these residues were mutated to alanines, the αIIbβ3 integrin could not be activated. With an antibody to phospho-S484, it was shown that kindlin-3 was phosphorylated in activated platelets and HEL megakaryocytic cells. In neutrophils, kindlin-3 phosphorylation was dependent on integrin-linked kinase binding to kindlin-3, and the phosphorylation was required for LFA-1 induced adhesion [31,32].

## 4. Integrin-Integrin Crosstalk Enables a Switch of Integrin Based Functions

The first indication that one integrin may dominate over another integrin came when LFA-1 was found to down regulate the α4β1 integrin in the binding of T lymphocytes to endothelial cells [33]. After that a large number of integrin-integrin crosstalks or transdominant regulations have been described [7], and only some can be mentioned here. When K562 erythroleukemia cells, which express the α5β1 fibronectin receptor, were transfected with the αVβ3 integrin, the α5β1-mediated phagocytosis of fibronectin-coated beads was inhibited [34]. Subsequent work showed that the β3-chain was sufficient and necessary to induce this function [35]. Likewise, transfected αIIbβ3 integrin inhibited the integrins α2β1 and α5β1 with their ligand fibronectin in Chinese hamster ovary (CHO) cells, and showed their requirement of the β3 domain for inhibition [36]. Porter and Hogg then showed that activation of LFA-1 in T cells resulted in strong inhibition of the α4β1 integrin and some inhibition of α5β1 [37]. Further work showed opposite crosstalk, where α4β1 interaction with VCAM-1 increased LFA-1 binding to ICAM-1, due to increased integrin avidity [38]. The α3β1 integrin, which binds to the α3 (IV) noncollagenous domain, was shown to inhibit αVβ3 integrin in renal papilla cells [39]. Crosstalk from one integrin can also strengthen the adhesion of another integrin. Once bound to fibronectin, the αV binding then induced additional binding sites of α5β1 to fibronectin [40]. A finding of potential clinical relevance was the discovery that monoclonal antibodies, reacting with LFA-1, may down regulate α4β1 [41].

An interesting observation, involving integrin crosstalk, was already done in 2010, when Steiner and co-workers found that under shear stress, T cell polarization and crawling were different on coated ICAM-1 or ICAM-2 than on VCAM-1 [42]. Under flow, T cells migrated upstream on ICAM-1, but downstream on VCAM-1 [43]. Then, Valignat and co-workers observed that whereas T cells under shear stress migrated upstream on coated ICAM-1, neutrophils migrated downstream [44]. These differences in migration were found to be due to crosstalk induced by LFA-1 on α4β1 [45]. When the flow ended, T cells, which had been exposed to shear stress, and had been migrating upstream on ICAM-1, began to migrate randomly. If the cells had been under shear stress on surfaces coated with both ICAM-1 and VCAM-1, they migrated upstream, but after the flow ended, the cells continued to migrate in the upstream direction [46]. This result shows that they had "migrational memory". The same group then studied how substrate stiffness influenced T cell mobility. They studied T cell migration on polyacrylamide gels of varying density, containing ICAM-1, VCAM-1 or a 1:1 mixture of these. Under static conditions, the cells showed an increase in mobility on ICAM-1 containing gels as a function of matrix stiffness, but not on the other substrates. The mechanosensitivity was overcome when α4β1 was blocked with soluble VCAM-1 [47]. The results show that T cells respond to matrix stiffness through LFA-1, and that the crosstalk between LFA-1 and α4β1 compensates for the changes in matrix stiffness. Under flow, the crosstalk did not affect the upstream migration on gels of varying stiffness.

## 5. Mechanisms of Integrin-Integrin Crosstalk

An integrin can induce changes in other integrin activity in a given cell by different mechanisms. These could include: (1) changes in integrin expression, (2) competition of integrin α-chains for the same β-chains, (3) competition between cytoplasmic adaptors for binding to different integrin cytoplasmic domains, (4) competition between adaptors for binding to common sites in the integrin cytoplasmic domains, (5) regulation by integrin phosphorylation, and (6) a combination of some of the above. These points are dealt with below.

### 5.1. Changes in Integrin Expression or Integrin Chain Availability

There are a few examples known where a change in the expression of an integrin polypeptide affects the cell surface expression of another integrin. Knockdown of the β3 chain in melanoma cells decreased the amount of αVβ3, but upregulated αVβ5 expression [7]. The surplus of αV formed a dimer with β5 when less β3 was present. Similar results had earlier been obtained with other cells [48]. Knockdown of β1 in mammary carcinoma cells increased the expression of β3 mRNA [49]. The results show that when blocking the expression of one integrin in cancer cells, one should be aware of the possibility of increased expression of other integrins, and the need of blocking them in order to inhibit metastasis.

### 5.2. Competition of Integrins for Cytoplasmic Adaptors

There is much evidence that several integrins use similar regulatory mechanisms and utilize common interactions with key cytosolic adaptors. Therefore, we anticipate that there must be competition between integrins for adaptors. Some cytoplasmic proteins important in cell adhesion have been studied extensively, and most of them bind directly to integrin cytoplasmic domains. Talin-1 and-2 are essential cytoplasmic components in integrin regulation and integrin crosstalk. The talins bind to two sites in integrin β-chains, the proximal NXXY/F motif and a site upstream of this (Figure 2). Mutations W739A, L746A and Y747A in recombinant β3 tail peptides inhibited talin binding [50]. The same mutations also attenuated transdominant inhibition of αIIbβ3. Another crucial protein for integrin regulation is kindlin. Three kindlin family members exist with different cellular distributions. Kindlin-1 is expressed mainly in epithelia, kindlin-2 is ubiquitously expressed and kindlin-3 is confined to hematopoietic cells. Genetic absence of kindlin-3 results in the LAD III syndrome, characterized by extensive bleeding and immunological defects [51,52,53]. The kindlins are involved in integrin- based adhesion, and cooperate with talin in adhesion. Kindlin-2 has been easiest to express and purify and it has been used in in vitro experiments. It binds to the T/S-NXXY/F sequence in the β-chains. There may exist competition between different integrin β-chains for kindlins, but a rather small amount of the normal levels of kindlins needed for functional adhesion could mean that the amount of kindlins is not a limiting factor [54]. FilaminA is inhibitory for adhesion and competes with other adaptors for binding to integrin β-chains (Figure 2) [55]. It is an important negative regulator of integrins, and when released from one integrin it can be used for binding to another integrin resulting in the down regulation of integrin activity.

### 5.3. Important Regulatory Motifs in the Integrin β-Chains for Integrin Activity and Crosstalk

It is becoming increasingly evident that the β-chain cytoplasmic motifs that are important in integrin signalling also have a pivotal role in integrin crosstalk. The requirements for integrin activity and trans-dominant inhibition have been studied in detail using the β3 cytoplasmic domain. CHO cells expressing the αIIbβ3 integrin were transfected with chimeric constructs of the IL-2 receptor (Tac) and β3. Expression of the full length Tac-β3 completely inhibited the binding of the ligand-mimetic and activation specific monoclonal antibody PAC-1 to αIIbβ3, and spreading and adhesion of the transfected cells on immobilized fibrinogen [50,56]. Transfection with a C-terminal RGT deleted construct and further deletions, and resulted in less adhesion and cell spreading. Deletions up to residue 754 strongly inhibited binding of soluble fibrinogen. Further deletions were inactive. The results show that residues covering the proximal NPLY sequence and the TST sequence are important for soluble fibrinogen binding, but the distal NITY sequence is less so (Figure 3).

The RGD sequence in integrin ligands is recognized by several β1 integrins, but also by αV integrins such as αVβ3 [57]. To study the requirement for crosstalk between β1 and β3 integrins, the β3 chain was first expressed in murine GEβ1 neuro-epithelial cells by retroviral transduction. Interestingly, β3/β1 cells showed less spreading on fibronectin than β1 cells, which meant that β1 inhibited β3 [58]. C-terminal deletions of β3 at β3/759, -756, -752 and -746 showed less cell adhesion on fibronectin. When the distal NITY sequence was deleted, there was significantly less spreading, and even less when the TST sequence was removed. The β3/751–759 sequence covers the kindlin binding site, but β3/S752 and β3/Y759 are also phosphorylation sites. Importantly, αVβ3 attenuated the RhoA G protein activity present in β1 cells, whereas Rac1 activity increased. RhoA is an important inducer of stress fibers, and Rac1 promotes actin branching.

### 5.4. Integrin Phosphorylation Provides a Means for Rapid and Specific Trigger to Induce Integrin Activity and Crosstalk

Motile cells must be able to rapidly activate adhesion, but also attenuate it. It would not be economical for the cells to synthesize and degrade or transport key adaptor proteins, or other important molecules taking part in fast adhesion cascades. Therefore, other means are necessary. Integrin phosphorylation has been studied extensively and it provides a trigger to induce rapid cell adhesion and signalling, in part by regulated binding of cytoplasmic interacting proteins. The topic has recently been reviewed [5]. Most studies have been done on leukocyte β2 integrins, but phosphorylation of β1 and β3 integrins have also been studied in detail. [59,60,61]. The known integrin phosphorylation sites are shown in Figure 4. It should be noted that although phosphorylation has been indicated in crosstalk, phosphorylation does not always result in crosstalk.

In β1 integrins, TT788/789 are important and their mutations to alanines inhibited cell adhesion [59,60,61]. The TT788/789DD mutation regained cell adhesion, talin and kindlin binding, indicating that the residues may be phosphorylated in wt cells [62]. Another research group could not verify all findings, because the phosphor-T788/789 antiserum was not specific [63]. S785 in β1 integrins is evidently important, because the S785D mutation, which to some degree mimics phosphorylation, promoted attachment, and inhibited the spreading and migration of transfected fibroblasts and teratocarcinoma cells [64].

The p21-activated kinase 4 (PAK4) bound to the SERS motif in the β5 integrin chain, and phosphorylated S759 and S762 [65] (Figure 4). This motif is upstream of the phosphorylation motifs in the other integrins. The phosphorylations increased αVβ5 cell adhesion and migration on vitronectin, but crosstalk to other integrins was not studied.

The α-chains of the β2 integrins are phosphorylated in resting cells on single serines: S1140 in αL, S1126 in αM and S1158 in αX [23,66,67]. In all cases, mutations to alanine inhibited adhesion. αD phosphorylation has not yet been studied. Inside-out activation of T cells through the T cell receptor or the SDF-1 chemokine receptor resulted in β2 phosphorylation on T758 [68]. In vitro studies have shown that several protein kinase C enzymes are able to phosphorylate the β2-chain [69]. Further studies of LFA-1, showed that the S1140A mutation inhibited the β2 phosphorylation on T758 [70]. Phosphorylation on T758 is functionally important. Dimeric 14-3-3 proteins bound to the phosphorylated residue, followed by binding of the G protein-exchange factor Tiam1, resulting in activation of the small G protein Rac-1 [71]. Simultaneously, filamin A was released from the β2-chain. Structural analysis showed that filaminA fits into the β2 chain pocket, but when β2 is phosphorylated on T758, there was no longer space for it, whereas 14-3-3ζ readily fitted into the pocket [72]. This signalling route is important for crosstalk from LFA-1 to α4β1.

Activation of LFA-1 was known to down regulate α4β1 [37] (Figure 5). We could confirm that α4β1-dependent adhesion and cell spreading to VCAM-1 were inhibited by active LFA-1 and αXβ2, whereas the S1140A mutated αL in LFA-1 inhibited crosstalk to α4β1 [73]. Paxillin binds to the α4β1 integrin through residues E983-Y991, and S988 on the α-chain is phosphorylated by PKA [74,75]. This phosphorylation regulated the association of paxillin with α4 [76]. Studies with Jurkat T cells showed that high affinity binding of paxillin occurred to the dephosphorylated α-chain, and it resulted in binding to VCAM-1. α4β1 was then able to stimulate LFA-1 and T cell migration [77]. α4β1 is phosphorylated on β1 (probably on T788/789) [59,63], and the α4β1 activity decreased in LFA-1 activated cells. Interestingly, the signalling from LFA-1 to α4β1 at least partially proceeds through the 14-3-3ζ/Tiam1/Rac-1 route. By transfection of Jurkat T cells with a T758 phosphorylated β2 peptide, blocking of α4β1 was likewise obtained, whereas the non-phosphorylated β2 peptide showed no effect. Inhibition of 14-3-3ζ binding to LFA-1, with the 14-3-3ζ blocking peptide R18, reduced the inhibition of α4β1. Similar results were obtained using an inhibitor of Tiam1. Rac-1 inhibition reduced cell adhesion and spreading in all cell lines [73].

LFA-1 crosstalk to α4β1 was further studied with monoclonal LFA-1 antibodies. LFA-1-activating antibodies, and those inhibitory antibodies which signal to α4β1 induced β2 T758 phosphorylation followed by 14-3-3 binding and signalling through Tiam1. Neutral LFA-1 antibodies showed no activity. The treatments abrogated T cell binding to VCAM-1 through α4β1, and decreased its phosphorylation [41]. More 14-3-3 bound to the β2-chain of LFA-1 in cells activated with the CBR LFA-1/2 antibody than to cells, which were treated with the non-activating TS2/4 antibody, whereas the opposite took place in α4 complexes. More talin was immunoprecipitated with α4 complexes from lysates treated with the CBR LFA-1/2 antibody than with the TS2/4 antibody. The intracellular signalling increased the phosphorylation of PLCβ3 on the Ser-1105 inhibitory site, and increased the phosphorylation of PLCγ1 on Tyr-783, indicating activation.

These results show that antibody treatments can result in unexpected effects, and the primary target molecule only acts as a receptor enabling downstream signalling and the activation of secondary molecules.

The migration of T cells on ICAM-1 induced spreading and an actin-rich leading edge, whereas on VCAM-1 they had a more elongated shape. These affects may be due to the Crk adaptor proteins, which are known to mediate actin-dependent cells migration induced by LFA-1 [78]. The LFA-1 ligation induced phosphoinositide 3-kinase and ERK pathways and phosphorylation of multiple kinases and adaptor proteins, whereas α4β1 engagement triggered remarkably less signalling events. It is possible that the differences between LFA-1 and α4β1 are due to crosstalk.

In endothelial cells, function blocking antibodies to β1 integrins inhibited the αVβ3 integrin. The treatments increased protein kinase A (PKA) activity and β3 phosphorylation on S752 [61]. Wild type and S752A mutated integrin bound strongly to a laminin ligand. However, the S752D mutated integrin did not. PKA phosphorylated inhibitor-1, which in turn inhibited protein phosphatase-1 (PP1). This resulted in increased phosphorylation of S752 and the blocking of adhesion. In Glanzmann’s thrombastenia, S752 is mutated to proline, and this prevents activation of αIIbβ3 and the consequent binding to fibrinogen and platelet aggregation. The mutation also inhibited the ability of αIIbβ3 to prevent ligand binding of the α2β1 integrin to collagen [61] by crosstalk. Talin is important in integrin crosstalk and PKA and PKC have been shown to regulate talin dependent crosstalk [79].

The β3 chain of αIIbβ3 binds the SH3 domain of Src tyrosine kinase through the β3 C-terminal RTE sequence, and Src is known to phosphorylate residues Y747 and Y759 in β3 [80]. Presently, we do not know how these phosphorylations affect the binding of adaptor proteins, and potentially crosstalk, but the Src binding site is close to the NPIY motif forming part of the kindlin binding site, and could affect kindlin binding [81].

The SARS-CoV-2 virus receptor ACE2 has a cytoplasmic tail, which shows similarities to the β3 integrin cytoplasmic domain (Figure 4) [5,10,11]. Therefore, it is possible that there exists crosstalk through phosphorylation switches between the integrins and the virus receptor. In addition, the virus spike protein contains an RGD motif, and recent work has shown that RGD antagonists inhibit virus binding to Vero E6 cells [14]. Lung epithelial cells expressing β1 integrins bound strongly to the virus, whereas a β1 knockout cell showed no binding [13]. These results show that integrins are involved in SARS-CoV-2 infection, and virus induced integrin crosstalk is highly possible.

There is relatively little information on the protein kinases and phosphatases that regulate integrin phosphorylation and crosstalk in addition to PKA, PKC and CaMK II [82]. Phorbol esters activate protein kinase C enzymes and strongly stimulate leukocyte adhesion. Therefore, purified PKCs were tested on the integrin β2 cytoplasmic peptide, and several PKCs were able to efficiently phosphorylate the substrate [69]. This indicates that they also phosphorylate integrins in vivo, but does not prove it. The AKT kinase has also been proposed to phosphorylate integrins [83]. The CaMKII kinase has been shown to associate with β1 in breast tumour cells [84]. Inhibitors of CaMKII prevent the increase in β1 T789 phosphorylation, which is due to Ndr1 kinase [85].

The PP1 protein phosphatase and the PP2A serine/threonine phosphatase have been implicated in integrin dephosphorylation [61,86], and recently, the PPM1F phosphatase was shown to dephosphorylate T788 in β1 [62]. T788 corresponds to T758 in β2 integrins, but whether the enzyme also acts on phosphorylated β2 is not known. The phosphatases may participate in the regulation of crosstalk. Table 1 shows a summary of some integrin crosstalks where the mechanisms have been studied.

## 6. Perspectives of the Role of Integrins in Crosstalk

Motile cells must be able to rapidly react to changes in their environment and adhere to their ligands expressed on cells or the extracellular matrix. This is certainly true for leukocytes and platelets. To get out from the circulation into tissues, leukocytes must be able to bind to endothelial cells, migrate through the endothelial layer, and interact with extracellular matrix molecules. This means that they must express adhesion molecules, which can change both the activity and specificity during their travel to the final targets. Protein phosphorylation is the major posttranslational modification of proteins, and it enables proteins to change their activity without the need of new protein synthesis and degradation.

Integrin-integrin crosstalk often occurs in pairs, and is probably more common with certain integrins. One much studied case is the crosstalk between LFA-1 and α4β1. This is understandable, because the integrins are often expressed in the same cells, and their functions are tightly connected. Another example is αIIbβ3 in platelets, and its crosstalk to α5β1 and α2β1.

The effects of antibody treatments on integrins inducing crosstalk must therefore be taken with caution. The antibody target may by crosstalk, signalling to another integrin, which results in secondary effects. This is observed with LFA-1 and α4β1 [41].

The presence of integrin crosstalk may enable the development of drugs that interfere with the crosstalks. In principle, they could be very specific and result in less side effects. Instead of wiping out all functions of a certain integrin, only a specific cell associated function would be affected. Presently, we are only at the beginning of an exciting development which during the coming years may change quite significantly.

## Figures and Tables

**Figure 1 cells-11-01685-f001:**
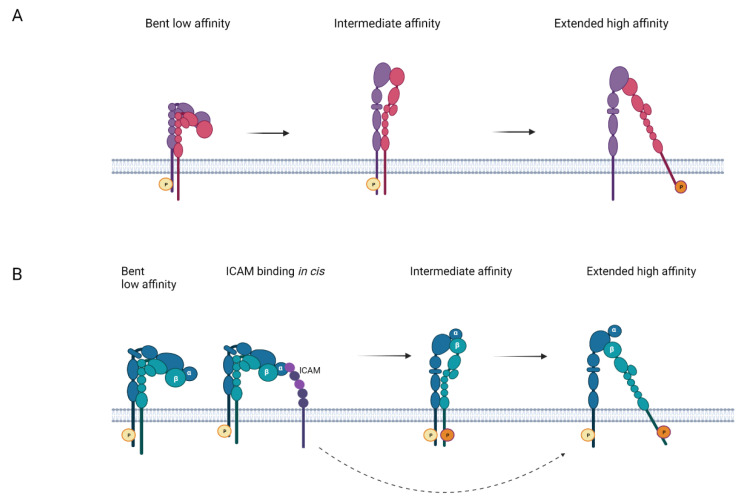
Schematic structures of resting and activated integrins. (**A**) an integrin with no I-domain, an example of which could be α4β1. In the resting integrin, the α-chain is phosphorylated, but not the β-chain. The α-chain in blue and the β-chain in red. (**B**), LFA-1, which contains an I-domain. Part of the molecules at the resting stage are bound to an ICAM in cis. The α-chain is phosphorylated on S1140. Initial activation results in the closed, extended form. The fully activated integrin is phosphorylated both on S1140 in the α-chain and T758 in the β-chain. P shows the phosphorylations.

**Figure 2 cells-11-01685-f002:**
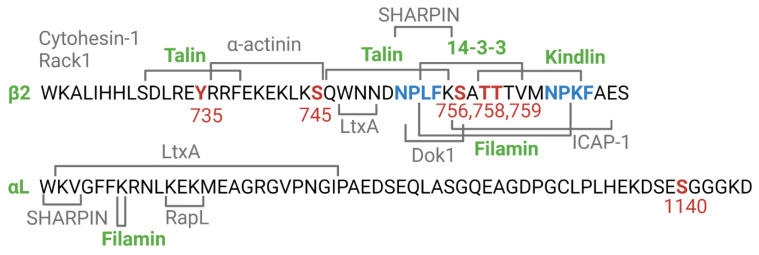
Binding sites of cytoplasmic proteins in LFA-1. The important NPXY/F motifs are in blue. The best studied proteins that are important in integrin crosstalk are coloured in green and the phosphorylation sites in red.

**Figure 3 cells-11-01685-f003:**
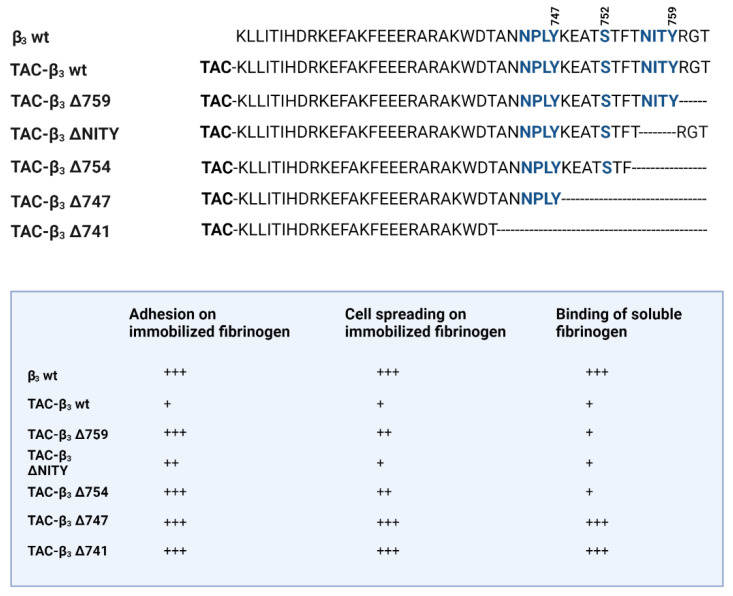
The cytoplasmic tail of the integrin β3 chain, and the effect of C-terminal deletions on the ability to induce trans-dominant inhibition of αIIbβ3 by the TAC-β3 integrin tail. The TAC-β3 construct completely inhibited adhesion and spreading on immobilized fibrinogen, and binding of soluble fibrinogen. The NITY deleted sequence also inhibited adhesion and spreading. C-terminal deletions up to T754 were able to inhibit the binding of soluble fibrinogen, but further shortenings were ineffective. The figure is adapted from ref. [56].

**Figure 4 cells-11-01685-f004:**
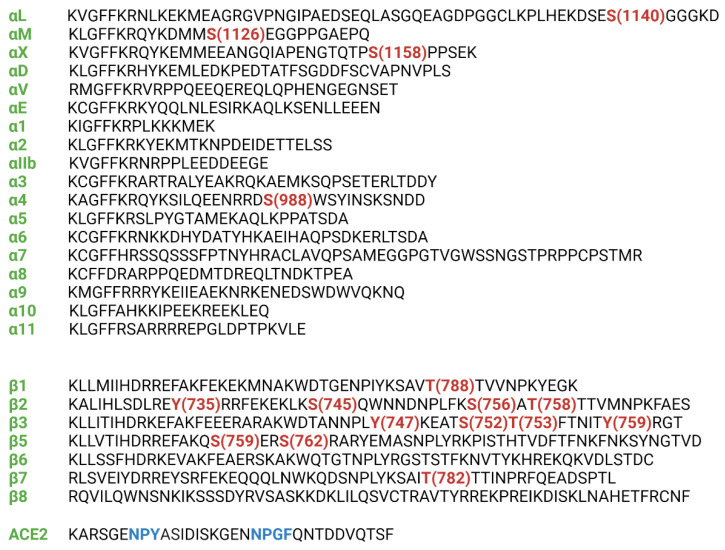
The sequences of the integrin and the major SARS-CoV-2 receptor ACE2 cytoplasmic tails. The known phosphorylation sites are marked in red. In ACE2, the NPXY/F like motifs are marked in blue.

**Figure 5 cells-11-01685-f005:**
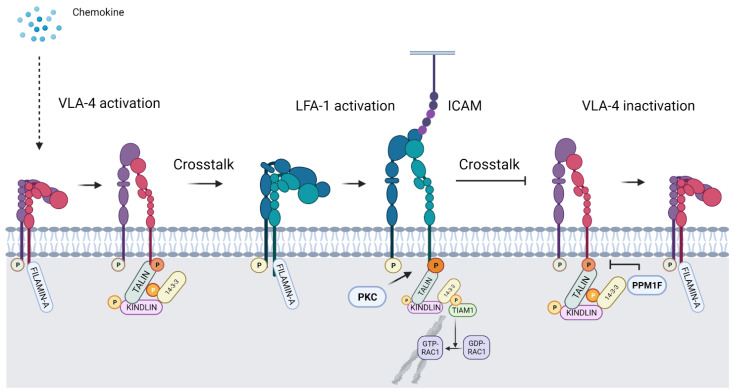
Crosstalk from VLA-4 to LFA-1, and from the activated LFA-1 to VLA-4. In the resting state VLA-4 (α4β1) is phosphorylated on the α-chain and filaminA is bound to the unphosphorylated β-chain. Upon activation, the β1-chain becomes phosphorylated, and through 14-3-3ζ/Tiam 1/Rac-1, the binding to the cytoskeleton is increased. Talin and kindlin-3 become associated with the β1-tail. LFA-1 is then activated and binds to an ICAM on a neighbouring cell, or crosstalks to an activated VLA-4 integrin, which loses the activation associated proteins and allows filaminA binding, resulting in inactivation.

**Table 1 cells-11-01685-t001:** Examples of integrin crosstalks.

Integrin Causing Crosstalk	Affected Integrin	Effect	Mechanism	References
α3β1, α6β1	αVβ3	INHIBITORY	PKA/PP1 β3 chain required β3 phospho-S752	[61]
αVβ3	α5β1	INHIBITORY	CAMKII β3 chain required β3 phospho-S752	[34,35,83]
αIIbβ3	α2β1, α5β1	INHIBITORY	β3 chain required β3 phospho-S752	[36]
αLβ2	α4β1	INHIBITORY	β2 chain required β2 phospho-T758	[37,41]
β3-integrin	α5β1	INHIBITION	Competition for talin	[50]
αLβ2	α5β1	INHIBITORY	Not known	[37]
α4β1	αLβ2	ACTIVATING	Not known	[38]
α3β1	αVβ3	INHIBITORY	Not known	[39]
αV-integrins	α5β1	ACTIVATING	Not known	[40]

## Data Availability

The figures were created using Biorender.com (accessed on 1 April 2022).

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
