# Peer review of "Regulation of Dynamic Cell Adhesion by Integrin-Integrin Crosstalk"

_cells, 2022, doi:10.3390/cells11101685_

Round 1

Reviewer 1 Report

This review summarized the studies on the regulation of integrin function by integrin-integrin cross talk. In general, this review is well-organized and summarized an interesting aspect of integrin research. There are few suggestions for the authors to further strengthen this review.

1.    Figure1 should contains both I-domain and I-less integrin because the authors are talking about a general concept of integrin activation in the text. It could make the readers confused if they only used LFA-1 as an example.
2.    Figure 2 showed some alpha subunit binding proteins. However, there is no description of these proteins in the text. Moreover, the authors have introduced alpha4 integrins in the review, some important regulatory proteins for alpha4 such as paxillin should be mentioned.  Again, the authors should reconsider whether it is suitable to use LFA-1 alone or include more integrin subunits that they have discussed in the review.
3.    A table that summarized the cross-talk between different integrins will be very helpful for the readers, which will strengthen the review.
4.    The key content of this review is the mechanism of integrin-integrin crosstalk, including: 1 ) changes in integrin expression, 2 ) competition of integrin α-chains for the same β-chains, 3 ) competition between cytoplasmic adaptors for binding to different integrin cytoplasmic domains, 4 ) competition between adaptors for binding to common sites in the integrin cytoplasmic domains, 5 ) regulation by integrin phosphorylation and 6 ) a combination of some of these. It would be better to discuss each topic point-by-point in detail.
5.    The authors declared that “the SARS-CoV-2 virus receptor ACE2 has a cytoplasmic tail, which shows similarities to the β3 integrin cytoplasmic domain”. However, Fig. 4 did not show the cytoplasmic tail of ACE2 and there was no description in the manuscript.

Author Response

Thank you for the expert comments of our manuscript Cells-1714988.

  1. We have made a new Fig. 1, A and B. In the A part a non-I domain integrin is depicted and in B LFA-1 as a well studied integrin. The figure text has been changed accordingly.
  2. This is somewhat problematic. Several integrins are known, they have different sequences and important sites. Some binding proteins have been studied with a certain integrin and others with another integrin. We show here LFA-1 as a well studied example. The paxillin binding site to the α4 chain is mentioned in the text.
  3. A Table 1 has been made with examples of integrin crosstalks, where the mechanism is at least partially understood.
  4. The key points have been dealt with under heading 5. Because the different integrin phosphorylations and adaptor protein bindings may occur simultaneously we did not keep the parts completely separated, but partially together under 5.1-5.4.
  5. The SARS-CoV-2 ACE2 receptor cytoplasmic sequence has been added to Fig. 4.

Reviewer 2 Report

This is an outstanding review; long waited for and very appropriate for the readership of this journal.

No edits needed.

Author Response

Thank you for your comments.